# Development of TiO_2_ Nanosheets with High Dye Degradation Performance by Regulating Crystal Growth

**DOI:** 10.3390/ma16031229

**Published:** 2023-01-31

**Authors:** Yasuyuki Kowaka, Kosuke Nozaki, Tomoyuki Mihara, Kimihiro Yamashita, Hiroyuki Miura, Zhenquan Tan, Satoshi Ohara

**Affiliations:** 1Graduate School of Medical and Dental Sciences, Tokyo Medical and Dental University, Bunkyo-ku, Tokyo 113-8549, Japan; 2State Key Laboratory of Fine Chemicals, School of Petroleum and Chemical Engineering, Dalian University of Technology, Panjin 124221, China; 3Joining and Welding Research Institute, Osaka University, 11-1 Mihogaoka, Ibaraki 567-0047, Osaka, Japan

**Keywords:** TiO_2_, nanosheet, facet engineering, photocatalytic activity

## Abstract

TiO_2_ nanosheets have been studied as photocatalysts in various fields, and their performance has been actively improved. Herein, we prepared titania nanosheets with a smaller size than those reported previously with a side length of 29 nm and investigated their photocatalytic activity. (NH_4_)_2_TiF_6_ and Ti(OBu)_4_ were used as raw materials, and the F/Ti ratio was varied in the range of 0.3 to 2.0 to produce a series of samples with different side lengths by hydrothermal synthesis. A reduction in the F/Ti ratio led to the reduced size of the titanium nanosheets. The photocatalytic activity of each sample was evaluated through the degradation of methylene blue (MB) under ultraviolet (UV) irradiation (365 nm, 2.5 mW/cm^2^). UV irradiation promoted the decomposition of MB, and the highest degradation efficiency was achieved using titania nanosheets prepared with a F/Ti ratio of 0.3. The high catalytic activity can be attributed to the increase in the surface area due to size reduction. The ratio of the {001} surface exposed on the titania nanosheet also affected the photocatalytic activity; it resulted in increased activation of the reaction. This study demonstrates that further activation of the photocatalytic activity can be achieved by adjusting the size of titania nanosheets.

## 1. Introduction

Titania (TiO_2_) is chemically and biologically inert, non-toxic, inexpensive, and environmentally safe, and has been studied as the most practical semiconducting catalyst owing to its excellent photocatalytic activity. It has wide-ranging applications in various fields [1]. In dentistry, titania is used as a denture cleanser, whitening material, and dental implant [2]. In addition, titania has been applied in environmental remediation [3]. Various studies have focused on improving its photocatalytic activity in the visible-light region [4,5]. However, the photocatalytic activity of visible-light-responsive titania remains unsatisfactory [6], and further studies are required to improve its activity.

One of the methods for improving the catalytic activity of titania is to change the facets of the titania surface. With regard to titania facet engineering, it has been reported that it is possible to increase the activity of titania by adjusting the ratio of the {001} and {101} facets of the titania surface [7,8,9]. It has also been reported that the functionality of titania nanosheets can be enhanced by inhibiting the recombination of electrons and holes generated under UV excitation [10,11]. Furthermore, it has been shown that the movement of electrons and holes affects the photocatalytic activity of titania because of the surface heterojunction between the {001} and {101} facets, and the photocatalytic activity changes depending on the facet ratio [7,12]. The photocatalytic effect of titania is due to redox reactions occurring on its surface using the photoinduced electrons (e^−^ and holes (h^+^). Several types of reactive species are generated by electrons and holes involved in the redox reaction. Oxygen is converted to reactive oxygen species (ROS) with high reactivity, such as superoxide and hydroxyl radicals, that are involved in the redox reaction. Therefore, it is important to strengthen the photocatalyst by controlling the ratio of the {001} and {101} facets and spatially separating the photogenerated electrons and holes [13]. The {001} facet with the higher surface energy compared to that of the {101} facet exhibits higher reactivity for the adsorption of reactants, retarding the growth of anatase TiO_2_ crystals along the {001} direction [14]. Upon using HF as a capping agent in the synthesis of titania, the crystal growth of the {101} facet is suppressed; however, this method has safety risks and wastewater issues due to residual HF [15]. Therefore, in the synthesis of TiO_2_, another material was used as the source of F ions, instead of HF, to ensure safety. Considering the safety of F ion supply, a previous study has used ammonium hexafluorotitanate as the F source in titania synthesis [16]. Titania nanomaterials, which have many exposed surfaces as a crystalline material and highly reactive facets that can be engineered for achieving desired crystallographic properties are promising for various applications [17,18,19].

The size of TiO_2_ nanoparticles (NPs) also plays a pivotal role in its performance as a photocatalyst [20]. Anpo et al. successfully fabricated TiO_2_ NPs with a wide range of sizes (3.8 to 200 nm) and showed that the bandgap decreased with a decrease in the number of NPs, but changed only slightly when the size was less than 12 nm [21]. Another study clarified that the optimum particle size for the reduction of CO_2_ is 14 nm after evaluating particles with side lengths of 4.5 to 29 nm [22]. Recently, various synthetic methods have been developed for producing TiO_2_ nanosheets with controlled sizes and morphologies [23,24]. Maisano et al. fabricated TiO_2_ nanosheets with side lengths in the range of 21.7 to 91 and reported that the nanosheet with a side length of 67.1 nm and {001} facet ratio of 59.3% exhibited the highest photoactivity [25]. We have also previously synthesized titania nanosheets in the size range of 29–550 nm by a hydrothermal method, in which the crystal growth of titania was effectively controlled by adjusting the F/Ti ratio. We found that titania nanosheets with a side length of 29 nm have the best catalytic activity [26]. However, the size effect of titania nanosheets with side lengths under 10 nm on their photocatalytic activity is not well known because of the difficulty in synthesizing small nanosheets.

Therefore, the objective of this study was to prepare nanometer-sized titania nanosheets with less than 10 nm side lengths and to clarify their functional properties. To evaluate the photocatalytic activities of titania samples with a controlled size in the range of a few nanometers to several hundred nanometers, a dye decolorization assay was conducted using methylene blue (MB) as a model dye. Furthermore, the ROS generated during the reaction were quantified using disodium terephthalic acid (DTA) and nitro blue tetrazolium (NBT). The results indicated that the photocatalytic activity of the titania nanosheet increases as its size decreases. This study provides new insights for the development of effective titania photocatalysts with controlled sizes.

## 2. Materials and Methods

### 2.1. Fabrication of TiO_2_ Nanosheets

Ammonium hexafluorotitanate ((NH_4_)_2_TiF_6_) and titanium butoxide (Ti(OBu)_4_, Sigma–Aldrich Co., St. Louis, MO, USA, 97%) were used as the raw materials for the synthesis of titania nanosheets [16]. (NH_4_)_2_TiF_6_ was dissolved in hydrochloric acid and then Ti(OBu)_4_ was added dropwise, and the resulting mixture was stirred for 2 h. Subsequently, the hydrothermal reaction of the precursor solution was conducted at 180 °C for 6 h. The product was ultrasonically washed with distilled water and methanol, and then freeze-dried using a freeze dryer (FDS-1000, TOKYO RIKAKIKAI CO., LTD., Tokyo, Japan). A series of titania nanosheet samples were prepared by varying the F/Ti ratio in the precursor mixture as 0.3, 0.5, 0.8, 1.0, 1.5, and 2.0 (the resulting samples are denoted as NS0.3, NS0.5, NS0.8, NS1.0, NS1.5, and NS2.0, respectively). The obtained powders were characterized by X-ray diffraction (XRD; D8 Advance, Bruker AXS GmbH, Karlsruhe, Germany), UV-visible (UV-vis) spectrophotometry (Jasco V-550, JASCO International Co., LTD., Tokyo, Japan), and transmission electron microscopy (TEM; H-7100, Hitachi High-Technologies Corporation, Tokyo, Japan). Furthermore, we analyzed the lattice parameters by the whole powder pattern decomposition (WPPD) method using analysis software (DIFFRAC.TOPAS version 4.2, Bruker AXS GmBH). Based on the length and thickness of titania measured from the TEM images, the ratio of the {001} facet exposed on the surface of titania was calculated using the following equations:(1)$$S001=2{\left(L-\frac{d}{tan\theta }\right)}^{2}$$
(2)$$S101=\frac{2d}{sin\theta }\left(2L-\frac{d}{tan\theta }\right)$$
(3)$$P001=\frac{S_{001}}{S_{001}+S_{101}}$$
where *L* is the average length of the titania nanosheet and *d* is the average thickness. S001 represents all {001} facets exposed in the single crystal and S101 represents the {101} facets. P001 is the percentage of the exposed {001} facet. *θ* = 68.3° is the theoretical value of the angle between the {001} and {101} facets of anatase [27].

The forbidden bandgap energy (*E*g) was calculated using the Tauc plot. The optical bandgap was estimated using the following equation:(4)$${\left(h\alpha \nu \right)}^{\frac{1}{n}}=k\left(h\nu -Eg\right)$$
where *h* is the Planck constant, *ν* is the frequency, α is the absorption coefficient, k is the proportionality constant, *E*g is the optical bandgap, and *n* depends on the type of transition in the semiconductor material, with *n* = 1/2 for direct allowable transitions. By verifying the linearity of the plot using this equation, the x-intercept of the line through the inflection point of the graph was estimated as the optical bandgap. Assuming that scattering is constant over the wavelength range used, the Kubelka–Munk function was used instead of the absorption coefficient of titania. The Kubelka–Munk transform of the following equation was applied to calculate the absorption coefficient by substituting the diffuse reflectance measurements [28]:(5)$$K=\frac{{\left(1-R\right)}^{2}}{2R}$$

Tauc plots were drawn based on the values obtained.

### 2.2. Dye Degradation Assay

To evaluate the photocatalytic activity of the titania nanosheets, a decolorization test was performed using a solution of MB (FUJIFILM Wako Pure Chemical Corporation, Osaka, Japan). In this study, commercially available titania NPs (titanium oxide (IV), anatase type, Fujifilm Wako Pure Chemical Industries, Ltd., Richmond, VA, USA) were used as the control. The titania sample was placed in a 48-well plate, and 2 mL of a 0.3 mM MB solution was mixed. The samples were irradiated with UV light at 20 °C for 1, 10, 30, 60, 120, 180, 240, and 480 min using a high-pressure mercury lamp (HL100G, Sen lights corporation) at 2.5 mW/cm^2^. The light intensity was measured using a UV intensity meter (UV-37SD, CUSTOM corporation, Tokyo, Japan). As a control group, the same static sample was shielded from light using aluminum foil and irradiated with UV light. After UV irradiation for a predetermined time, a known amount of the sample was collected and centrifuged at 13,000 rpm for 10 min. The supernatant was diluted 10-fold, and the absorbance of the diluted solution at 630 nm was measured using a microplate reader (Model680, Bio-Rad Laboratories Inc., Hercules, Berkeley, CA, USA).

### 2.3. Chemical Analysis of ROS Formation in the Presence of TiO_2_

Solutions of DTA (Tokyo Chemical Industry Co. Ltd., Tokyo, Japan) and NBT (Tokyo Chemical Industry Co. Ltd., Tokyo, Japan) were prepared to evaluate the ROS emitted from TiO_2_ and their reactions under UV irradiation. The NBT solution (1 mmol) with 4 mg/mL TiO_2_ was irradiated for 2 h at 365 nm (power density: 2.5 mW/cm^2^). After irradiation, the samples were centrifuged at 13,000 rpm for 10 min, the supernatant was collected, and 100 μL of dimethyl sulfoxide (DMSO; FUJIFILM Wako Pure Chemical Corporation, Osaka, Japan) was added and stirred for 10 min. The absorbance was measured at 570 nm using a UV-vis spectrophotometer.

A DTA solution (20 mmol) was added to TiO_2_ and centrifuged in the same manner as described above, and the fluorescence intensity (Ex 340/Em 460 nm) of the collected supernatant was measured using a fluorescence microplate reader (Wallac Arvo MX, Perkin Elmer Co., Ltd., Waltham, MA, USA).

The generated hydroxyl radicals and superoxide were estimated using calibration curves, as reported previously [26]. Briefly, standard solutions of 2-hydroxy terephthalic acid and formazan were prepared at representative concentrations, and the fluorescence intensity and absorbance at 580 nm were measured.

### 2.4. Statistical Analysis

Statistical analysis was performed using the one-way analysis of variance with Bonferroni post hoc tests for the changes in the absorbance of MB under UV irradiation and the differences in superoxide production, and Dunnett’s T3 correction was used for the data of MB degradation in the dark and hydroxyl radical production using SPSS (IBM SPSS Statistics, version 27, Armonk, NY, USA). *p*-values less than 0.05 were considered statistically significant.

## 3. Results

### 3.1. Characterization of TiO_2_ Nanosheets

The XRD patterns of the prepared TiO_2_ powders are shown in Figure 1a. The patterns of all samples contained peaks consistent with anatase-type titania (PDF# 21-1272). In addition, an increase in the peak width, including for those at 25, 48, 54, and 63°, was observed with a decrease in the F/Ti ratio in the precursor mixture. Lattice parameters of TiO_2_ nanosheet were analyzed by WPPD method and are shown in Figure 1b. The F/Ti ratio had no effect on the lattice parameters of *a*-axis and *c*-axis.

The TEM images of the prepared TiO_2_ nanosheets are shown in Figure 2a. The length and thickness of the titania nanosheets were measured from the TEM images (*n* = 6). The length and thickness decreased as the F/Ti ratio decreased (Figure 2b). The average lengths and thicknesses of the TiO_2_ nanosheets were determined to be 6.3 and 4.9 nm (NS0.3), 14 and 8.3 nm (NS0.5), 21 and 9.5 nm (NS0.8), 34 and 13 nm (NS1.0), 69 and 15 nm (NS1.5), and 445 and 64 nm (NS2.0), respectively. The {001} facet ratio increased with increasing F/Ti ratio, and the smallest {001} facet ratio was obtained for NS0.3 (Figure 2c). Further, Tauc plots were drawn from the UV–vis absorbance results (Figure 3), and the optical bandgaps were determined using Equation (5). The results are shown in Table 1.

### 3.2. Assay of Dye Degradation by TiO_2_ Nanosheets

When the absorbance of the MB solutions added with the TiO_2_ nanosheets was monitored, it was clear that the absorbance of each solution decreased over time, indicating that the decomposition of MB was promoted. The results of the degradation assays are presented in Figure 4. When the UV-irradiated sample was compared with the light-shielded one, the UV-irradiated solution was strongly decolorized. After 480 min, NS0.3 provided the best results, followed by NS0.5, NS1.0, NS0.8, NS1.5, NS2.0, and titania NPs, and the decomposition efficiency decreased in that order (Figure 5a). After 480 min of UV irradiation, the calculated rate constants were 0.0007 min^−1^ for NS0.3 and NS0.5; 0.0006 min^−1^ for NS0.8 and NS1.0; 0.0005 min^−1^ for NS1.5; and 0.0002 min^−1^ for NS2.0 and NP (Figure 6a). On the other hand, the absorbance of the MB solution with the nanosheets and nanoparticles decreased even without UV irradiation during 120 min, and NS0.3 showed a high decolorization ability at 120 min without irradiation (Figure 5b and Figure 6b).

### 3.3. Comparison of ROS Generation on TiO_2_ Nanosheets under UV Irradiation

Chemical analysis was performed using DTA and NBT to detect the active oxygen species generated by TiO_2_ during UV irradiation at 365 nm (2.5 mW/cm^2^) for 2 h at 20 °C. A mixture of each TiO_2_ with a different size and DTA was used to detect hydroxyl radicals, while a mixture with NBT was used to detect the changes in superoxide anion formation under UV irradiation. The results of these experiments are shown in Figure 7. Among the titania nanosheet samples, NS1.5 produced the highest number of hydroxyl radicals, followed by NS0.8 and NS1.0. Conversely, NS0.3 and NS0.5 produced fewer hydroxyl radicals than the others. Furthermore, NS1.0 produced the highest amount of superoxide. However, the difference in the amounts of superoxide produced by different nanosheet samples was not significant. Commercially available titania NPs from Wako were used as control.

## 4. Discussion

Considering the safety and wastewater issues, in this study we used ammonium hexafluorotitanate as the source of F ions instead of HF and successfully prepared highly functional titania nanosheets with a size as small as 6.3 nm by hydrothermal synthesis. We varied the F/Ti ratio of the precursor solution from 0.3 to 2.0 to obtain a series of samples to study the size effect of the titania nanosheet on its photocatalytic activity. TEM was used to clarify the structure and XRD analysis was used to study the crystal phase of the prepared titania. Moreover, the crystal type was identified by XRD, and the optical bandgaps were determined using Tauc Plots. Decolorization experiments performed under UV irradiation and in the dark using MB as the model dye to investigate their photocatalytic activities revealed that NS0.3 has the strongest decolorizing power under UV irradiation. NS0.3 was effective even in the dark. To further investigate the photocatalytic function of titania, experiments were conducted to measure the amount of ROS produced under UV irradiation. DTA and NBT were employedto determine the amounts of superoxide anions and hydroxyl radicals, respectively.

TEM images indicated that the formation of the {001} plane was suppressed as the F/Ti ratio increased, although the formation of the {101} was enhanced. This result indicates that ammonium hexafluorotitanate plays an important role in the crystal growth of titania. In a previous study, the {001} facet ratio of TiO_2_ nanoparticles with a side length of 13 nm synthesized without capping agents was found to be 0.11 [12]. However, our study reveals that ammonium hexafluorotitanate can generate a smaller titania nanosheet with a side length of 6.3 nm with a higher {001} facet ratio of 0.4.

Our study also indicated that the optical bandgap of the titania nanosheet decreases from 3.29 to 3.20 eV as the side length decreases from 256 to 6.7 nm. The optical bandgap of titania NPs decreases with decreasing particle size [29]. Furthermore, the surface of TiO_2_ nanosheet synthesized using fluoride ion as the capping agent has been reported to be covered with fluoride, resulting in the generation of Ti_(1−x)_^4+^Ti_x_^3+^O_(2−x)_^2−^F_x_^−^ on the nanosheet [30]. The formation of Ti^3+^ resulted in excess unpaired electrons, and the reduced state is relatively close to the bottom of the conduction band. The results of the present study are consistent with those of the previous reports, indicating that the optical bandgaps can be determined not only by the energy state of ions but also by the crystal size and shape.

An MB degradation assay and measurement of hydroxyl radical and superoxide anion formation were performed to evaluate the photocatalytic activities of the prepared titania nanosheets. According to the MB degradation assay, NS0.3 was found to have the highest decolorization ability under UV irradiation because of the improvement in its reactivity owing to its high surface area and the activation effect of the {001} and {101} surfaces. Furthermore, NS0.3 exhibited better decolorization performance even under dark conditions. This is because the decolorization process involves two steps: dye adsorption on the titania surface and its subsequent degradation via photocatalysis [31]. The variation of the ln(A0/A) of MB with its reaction time in the dark clearly indicated that the adsorption of the dye continued until 120 min; the adsorption ability of NS0.3 was the best among the samples.

The quantification of ROS is important for understanding the mechanism of photocatalysis. Since hydroxyl radicals and superoxide anions play critical roles in dye degradation via redox reactions [32], the degradation pathway of MB includes the degradation of chromophoric and auxochrome groups by the generated hydroxyl radicals and superoxide anions [33,34]. TiO_2_ suspensions can induce the formation of hydroxyl radicals from H_2_O and superoxide anions from oxygen dissolved in water under UV irradiation. NS0.8, NS1.0, and NS1.5 produced higher quantities of hydroxyl radicals, while NS0.3 and 0.5 produced less. Superoxide anions were produced most abundantly by NS1.0. It has been reported that the formation of hydroxyl radicals and superoxide anions is highly dependent on the {101} and {001} facets of the heterojunction of TiO_2_ nanosheets [12]. Although some groups have reported that the {101} facet is more active than the {001} facet [35], others have reported that the {001} facet is more active than the {101} facet when the activation of the plane is considered by taking the titania size into account [36]. However, it has been shown that an appropriate ratio of the {101} facet to {001} facet may increase the activity of titania [12]. Accordingly, the facet ratio of NS1.0 was expected to be optimal for ROS formation. However, NS0.3 exhibited the best performance in MB decolorization. This is because, despite the decrease in the number of {001} facets, the adsorption of dye increases with decreasing size. This result indicates that the size reduction of titania has a significant effect on the improvement of the decolorization performance.

The present method facilitates the fabrication of nanometer-sized titania nanosheets, and NS0.3 was found to have the highest photocatalytic activity, which is not simply because of the size reduction of titania. In this study, the particle size was controlled by varying the F/Ti molar ratio in the range of 0.3 to 2. At a F/Ti molar ratio below 0.3, the expected titania nanosheets could not be isolated. It has also been reported that above a F/Ti molar ratio of two, excess fluoride ions enter the lattice and inhibit the formation of TiO_2_ with exposed {001} facets [17]. Our results indicate that the balance between the amount of ROS produced and the increase in the surface area due to size reduction affects the photocatalysis of TiO_2_ nanosheets. Further studies are required to clarify the electron–hole pair excitations in single nanometer-sized titania NS using an electron paramagnetic resonance.

## 5. Conclusions

The size of the titania nanosheets was controlled by adjusting the titanium-to-fluorine ratio in the precursor solution, and a decolorization assay was conducted using MB as a model dye. TiO_2_ nanosheets (NS0.3) with the minimum size were obtained at a F/Ti ratio of 0.3. Commercially available titania NPs (titanium (IV) oxide, anatase type) from Fujifilm Wako Pure Chemical Industries, Ltd. were used as the reference sample.

The results of the study are as follows:The dye was decolorized more strongly under UV irradiation than in the dark;NS0.3 had the best decolorizing power, followed by NS0.5, NS1.0, NS0.8, NS1.5, and NS2.0.

## Figures and Tables

**Figure 1 materials-16-01229-f001:**
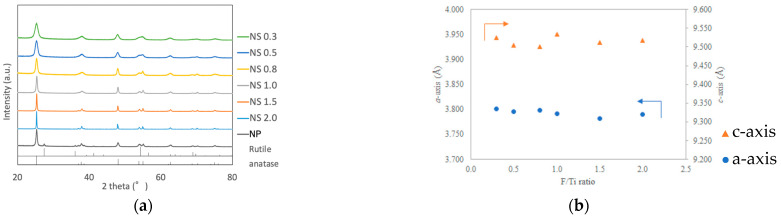
Characterization of TiO_2_ nanosheets. (**a**) XRD patterns of TiO_2_ NS0.3, NS0.5, NS0.8, NS1.0, NS1.5, NS2.0, and TiO_2_ nanoparticles (NPs). (**b**) Lattice parameters of TiO_2_ nanosheet.

**Figure 2 materials-16-01229-f002:**
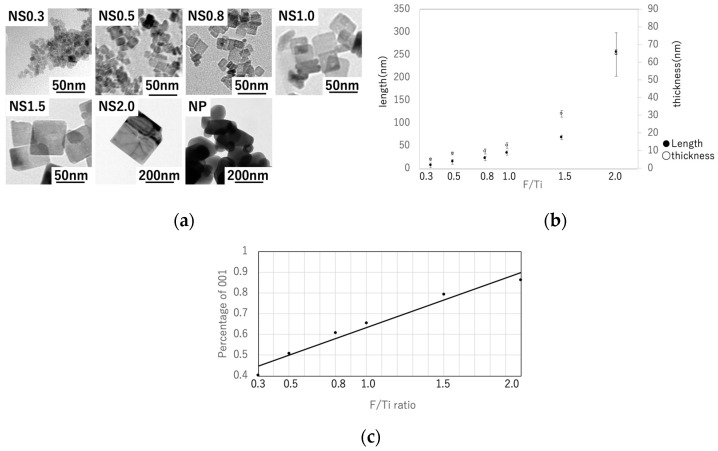
(**a**) TEM images of various TiO_2_ nanosheets and NPs. (**b**) Length and thickness of each TiO_2_ sample measured from TEM images. (**c**) Ratio of the {001} facet (P_001_) calculated from the length and thickness.

**Figure 3 materials-16-01229-f003:**
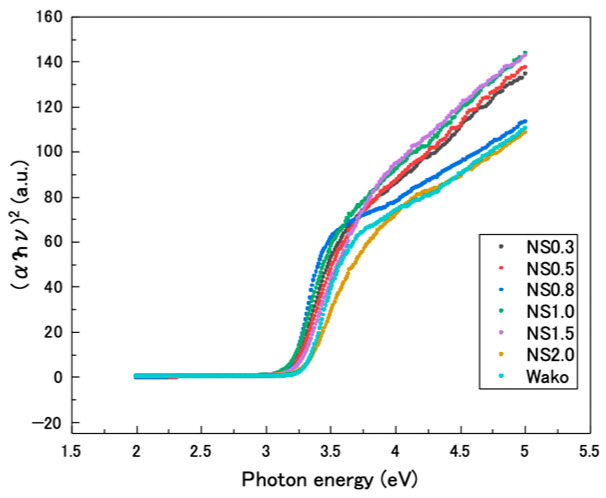
Tauc plots of TiO_2_ nanosheet and TiO_2_ nanoparticle samples.

**Figure 4 materials-16-01229-f004:**
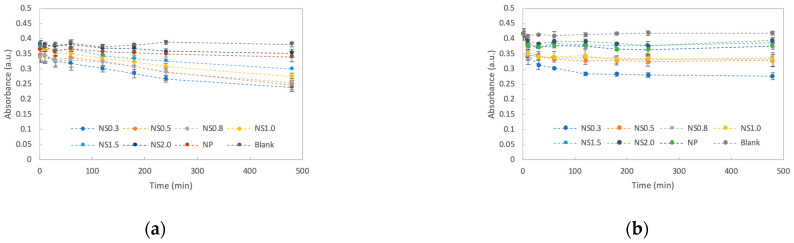
Time course of the decolorization of methylene blue in the presence of prepared titania samples. (**a**) MB decolorization under UV irradiation. (**b**) MB decolorization in the dark.

**Figure 5 materials-16-01229-f005:**
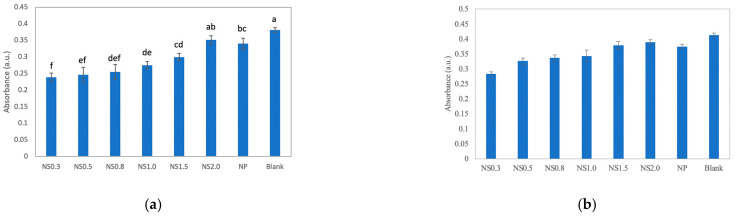
(**a**) Multiple comparisons of the MB absorbance under UV irradiation after 480 min. (**b**) Multiple comparisons of the MB absorbance in the dark after 120 min. The same letter indicates no significant difference at α = 0.05.

**Figure 6 materials-16-01229-f006:**
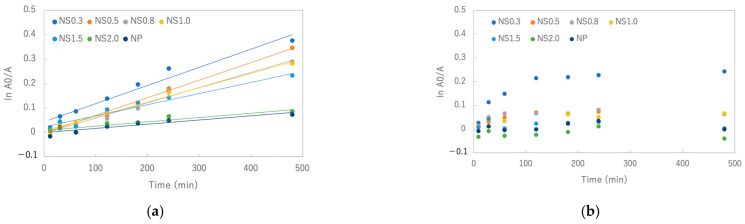
Variation of ln(Ao/A) of MB with reaction time (**a**) under UV irradiation and (**b**) in the dark.

**Figure 7 materials-16-01229-f007:**
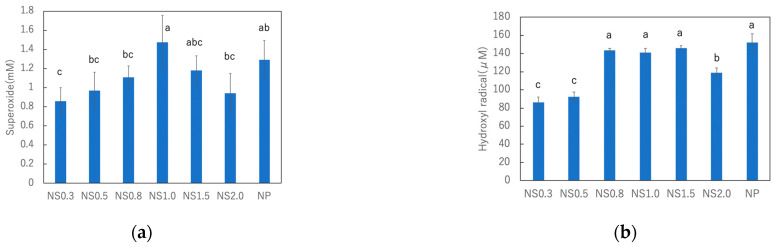
Amount of hydroxyl radicals (**a**) and superoxide anions (**b**) generated under UV irradiation by titania nanosheets and nanoparticles. The same letters are not significantly different at α = 0.05.

**Table 1 materials-16-01229-t001:** Optical band gaps of various TiO_2_ samples.

TiO_2_	Optical Bandgap (eV)
NS0.3	3.2
NS0.5	3.22
NS0.8	3.2
NS1.0	3.2
NS1.5	3.24
NS2.0	3.29
NP (control sample)	3.29

## Data Availability

Not applicable.

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
