# Peer review of "Development of TiO2 Nanosheets with High Dye Degradation Performance by Regulating Crystal Growth"

_materials, 2023, doi:10.3390/ma16031229_

Round 1
Reviewer 1 Report
In this manuscript, Kowaka et al. described the “Development of TiO2 nanosheets with high dye degradation performance by regulating crystal growth” in detail. All the samples are characterized adequately and the activities are good. However, at this stage there are still many problems and I therefore, suggest major review for this manuscript keeping in mind the following questions.
1) Why the introduction of F reduced the surface area significantly? How does F-atoms replace O in the nanostructure. This must be explained in detail.
1) Do you have introduced new ideas to overcome the limitations of different research papers as you mentioned in the abstract as “their conclusions are not always consistent, due to the variety of applied experimental conditions and the different solid materials used as catalysts”.
2) What is the reason behind decreased length and thickness of TiO2 after the introduction of F?
3) What is the light intensity of the used for irradiation?
4) The TEM images show particle morphology and not nanosheets morphology. Can you take more images to support your statement.
5) The effect of F-doping on the XRD patterns must be explained in detail.
6) English language of the paper needs careful attention to attract the readers.
7) Did the author perform any test for the measurement of the intermediates formed during the decomposition of the dyes?
8) The introduction of F-decreased the band gap although the atomic orbitals of F are at lower energy state than oxygen. Through a bird eye view on it.
9) No specific experiment has been conducted for charge separation which plays major role in the overall photocatalytic redox reactions.
10) Some very important citations are missing.
i) A. Zada, M. Khan, Z. Hussain, M. I. A. Shah, M. Ateeq, M. Ullah, N. Ali, S. Shaheen, H. Yasmeen, S. N. A. Shah, A. Dang, Extended visible light driven photocatalytic hydrogen generation by electron induction from g-C3N4 nanosheets to ZnO through the proper heterojunction, Z. Phys. Chem. 236 (2022) 53-66.
ii) A. Zada, N. Ali, F. Subhan, N. Anwar, M. I. A. Shah, M. Ateeq, Z. Hussain, K. Zaman, M. Khan, Suitable energy platform significantly improves charge separation of g-C3N4 for CO2 reduction and pollutant oxidation under visible-light, Prog. Nat. Sci. Mat. Int. 29 (2019) 138-144.
Reviewer 2 Report
This paper reports the structure and photo-catalytic performance of TiO 2 nanosheet. The structure tests should be improved.For example, the high resolution of TEM should be provided. On the other hand, it is best to provide the XPS. The electronic state of F should be provided to clarify the state of the F atoms. The photocatalytic performance of the F-doped TiO2 should be enhanced. At present, the performance is not better enough for publication.
Reviewer 3 Report
Article “Development of TiO2 nanosheets with high dye degradation performance by regulating crystal growth” by Yasuyuki Kowaka, Kosuke Nozaki et al. is devoted to synthesis of the titania nanosheets and investigation of their photocatalytic activity through the degradation of methylene blue under UV irradiation. The article describes some interesting research evidence, but below are some of the reviewer's comments.
The Abstract should indicate the size of TiO2 nanoparticle instead of referring to previous studies.
In the Introduction Sources of literature should be supplemented with more recent articles and reviews related to the production, research and use of titanium dioxide with modern works. Some references (3, 4, 13, 17, etc.) are designed carelessly and non-standard. Most of the References are outdated (released more than 5 years ago), so the authors should revise the Manuscript taking into account actual researches.
Row 35. What do the authors mean by "In addition, titania has been applied in environmental remediation [3]"? The cited link from 2005 is completely inconsistent with the statement.
Row 170. Figure 1a for XRD.
“…an increase in the peak width…”. It is worth clarifying which peak changed.
It should be noted that the presented TEM images show rather strong shape anisotropy of titania nanoparticles, especially for samples NS 1.5 and NS 2.0. This should be reflected in the study. The shape of sample NS 0.5, according to the imagine, resembles a spherical one.
Row 214. NP is a not entirely clear abbreviation in the Table. It should be clarified that this is a comparison sample.
What do the letters a, b, c and others in Fig. 4a stand for?
Row 284. “To investigate the structure of each of the prepared titania nanosheets, the sizes and number of {001} and {101} facets were determined through TEM and XRD analyses.” It should be noted that the 001 and 101 faces are not defined in any way on the diffraction patterns or on the TEM images. The number of faces, according to the given data, was determined only by processing the TEM photographs. XRD was used to confirm the formation of the anatase phase for all synthesized samples.
Row 278 is repeated with Row 290 (about the source of ions F-).
Round 2
Reviewer 1 Report
The respected authors have removed all the mistakes/shortcomings and improved the quality of the paper significantly, therefore, I recommend the publication of "Development of TiO2 nanosheets with high dye degradation performance by regulating crystal growth" in your reputed journal.
Reviewer 2 Report
The paper has been revised according to reviewer's comments. It can be accepted now.